



# Carbonaceous aerosol source apportionment using the aethalometer model-Evaluation by radiocarbon and levoglucosan analysis at a rural background site in southern Sweden

Johan Martinsson[1,2], Hafiz Abdul Azeem[3], Moa K. Sporre[4], Robert Bergström[5,6], Erik Ahlberg[1,2],
Emilie Öström[1,2], Adam Kristensson[1], Erik Swietlicki[1], Kristina Eriksson Stenström[1]

[1]Division of Nuclear Physics, Lund University, Box 118, SE-22100, Lund, Sweden
[2]Centre for Environmental and Climate Research, Lund University, Ecology Building, SE-22362, Lund, Sweden
[3]Centre for Analysis and Synthesis, Department of Chemistry, Lund University, Lund, Box 118, SE-22100, Lund, Sweden
[4]Department of Geosciences, University of Oslo, Postboks 1022, Blindern, 0315, Oslo, Norway
[5]Swedish Meteorological and Hydrological Institute, SE-60176, Norrköping, Sweden
[6]Department of Chemistry, University of Gothenburg, SE-41296, Gothenburg, Sweden

*Correspondence to*: Johan Martinsson (johan.martinsson@nuclear.lu.se)

**Abstract.** With the present demand on fast and inexpensive aerosol source apportionment methods, the aethalometer model
was evaluated for a full seasonal cycle (June 2014-June 2015) at a rural atmospheric measurement station in southern
Sweden by using radiocarbon and levoglucosan measurements. By utilizing differences in absorption of UV and IR, the
aethalometer model apportions carbon mass into wood burning (WB) and fossil fuel combustion (FF) aerosol. In this study, a
small modification in the model in conjunction with carbon measurements from thermal-optical analysis allowed
apportioned non-light absorbing biogenic aerosol to vary in time. The absorption differences between WB and FF can be
quantified by the absorption Ångström exponent (AAE). In this study $AAE_{WB}$ was set to 1.81 and $AAE_{FF}$ to 1.0. Our
observations show that AAE was elevated during winter (1.36±0.07) compared to summer (1.12±0.07). Quantified WB
aerosol showed good agreement with levoglucosan concentrations, both in terms of correlation ($R^2$=0.70) and in comparison
to reference emission inventories. WB aerosol showed strong seasonal variation with high concentrations during winter (0.65
µg m$^{-3}$, 56 % of total carbon) and low concentrations during summer (0.07 µg m$^{-3}$, 6 % of total carbon). FF aerosol showed
less seasonal dependence, however black carbon (BC) FF showed clear diurnal patterns corresponding to traffic rush hour
peaks. The presumed non-light absorbing biogenic carbonaceous aerosol concentration was high during summer (1.04 µg m$^{-3}$, 72 % of total carbon) and low during winter (0.13 µg m$^{-3}$, 8 % of total carbon). Aethalometer model results were further
compared to radiocarbon and levoglucosan source apportionment results. The comparison displayed good agreement in
apportioned mass of WB and biogenic carbonaceous aerosol but discrepancies were found for FF aerosol mass. The
aethalometer model overestimated FF aerosol mass by a factor of 1.3 compared to radiocarbon and levoglucosan source
apportionment. This discrepancy may possibly be explained by a relatively low $R^2$ value in the fit of FF aerosol light
absorption to carbon mass concentration. In summary, the aethalometer model offers a cost-effective, yet robust high-time
resolution source apportionment at rural background stations compared to a radiocarbon and levoglucosan alternative.





## 1 Introduction

Carbonaceous aerosol, i.e. the fraction of the aerosol containing carbon, is approximately contributing with 25 % to the mass of particulate matter with smaller diameter than 10 μm ($PM_{10}$) in Europe (Fuzzi et al., 2015) and is presently estimated to be one of the most important climate forcers (IPCC, 2013). However, the magnitude of carbonaceous aerosol impact on climate

is still associated with significant uncertainty (IPCC, 2013). The carbonaceous aerosol originates mainly from three sources; wood burning, fossil fuel combustion and biogenic emissions. Black carbon (BC) or soot is formed from incomplete combustion of fossil fuels and biofuels. BC has a graphitic carbon structure and is known to efficiently absorb incoming solar radiation (Bond et al., 2013). This absorption leads to molecular vibration and rotation which causes emission of longwave radiation, heating the atmosphere. On the other hand, the organic aerosol (OA) is known to mainly scatter

incoming sunlight, thereby cooling the climate. Recently, the strongly ultraviolet-absorbing brown carbon (BrC) has gained interest in the scientific community (Laskin et al., 2015;Martinsson et al., 2015;Saleh et al., 2013;Saleh et al., 2014). BrC is emitted in large quantities from wood burning and has been proposed to affect lower tropospheric photochemistry by reducing ultraviolet (UV) radiation (Jacobson, 1999). Although BrC is a much less effective light absorber than BC, deposition of BrC on bright surfaces such as snow or ice may cause significant changes in albedo (Doherty et al., 2010).

Carbonaceous aerosols have also been linked to serious health effects, mainly through inhalation (Grahame et al., 2014;Laden et al., 2006;Pope and Dockery, 2006). Carbonaceous aerosols derived from wood burning have been shown to be hazardous to humans (Barregard et al., 2006;Eriksson et al., 2014;Jalava et al., 2010;Naeher et al., 2007;Sehlstedt et al., 2010;Unosson et al., 2013). Additionally, diesel and gasoline vehicles emit large quantities of BC and associated compounds (e.g. polycyclic aromatic hydrocarbons, PAH) which have been suggested as one of the most health-damaging particle types

(Benbrahim-Tallaa et al., 2012;Hoek et al., 2002;Salvi et al., 1999).

One of the re-emerging air pollutants in Europe is particles from wood burning. Wood burning is increasing with approximately 3.5 % per year in Europe due to its potential $CO_2$-neutral effect on climate, while the fossil energy consumption is decreasing by 2 % per year (EEA, 2015). Particle emissions from wood burning are usually elevated during winter. It has been estimated that 45-65 % of the total ambient carbonaceous aerosol mass (TC) in Europe is associated with

wood burning during this period of the year (Gilardoni et al., 2011;Szidat et al., 2006). Due to the severe climate and health effects from different particle sources, and the importance of wood burning in particular, it is crucial to develop and evaluate source apportionment methods of the carbonaceous aerosols. An accurate source apportionment enables justified mitigation of particle emissions that affect health and climate, as well as a possibility to evaluate emission inventories and chemical transport models.

Levoglucosan is an anhydrosugar formed during pyrolysis of cellulose at temperatures above 300 °C (Simoneit, 2002). Due to its specificity for cellulose combustion, it has been widely used as a molecular tracer for wood burning in source apportionment studies (Gelencser et al., 2007;Genberg et al., 2011;Yttri et al., 2011a;Yttri et al., 2011b). However, there are some drawbacks of using levoglucosan for this purpose. Several studies have shown that levoglucosan may not be stable in



the troposphere, it may react with OH both in the gas-phase (Hennigan et al., 2010;May et al., 2012) and aqueous phase (Hoffmann et al., 2010;Zhao et al., 2014) leading to relatively short estimated atmospheric life-times of 1-5 days, depending on the season and atmospheric conditions. The importance of the degradation of levoglucosan in the ambient atmospheric aerosol is still not clarified (Yttri et al., 2015). Also, the relative levoglucosan contribution to the carbonaceous aerosol mass

is dependent on combustion conditions (Hedberg and Johansson, 2006). Levoglucosan is most commonly measured on aerosol-laden filters. Filter sampling is generally associated with low time resolution which makes it difficult to study rapid variations of this source marker.

More recently, the aethalometer model (Sandradewi et al., 2008a), employing multi-wavelength light absorbing measurement techniques with high time resolution, has been used for a number source apportionment studies (Favez et al.,

2009;Favez et al., 2010;Herich et al., 2011;Sandradewi et al., 2008a), as an alternative to the methods based on chemical analysis of filter samples. This method relies on the assumption that particles generated from wood burning are relatively more light-absorbing in the UV than infrared (IR) compared to particles from traffic and other fossil fuel combustion (Kirchstetter et al., 2004). The difference in light absorption can be quantified using the absorption Ångström exponent (AAE) which is a measure of the spectral absorption dependence (Kirchstetter et al., 2004). Wood burning emissions are

assumed to have an AAE between 1.5-2.5 while traffic and fossil fuel combustion derived particles exhibits an AAE around 1.0 (Kirchstetter et al., 2004). Despite the great benefits the light-absorption based source apportionment can offer, with its high time resolution and low costs, the relations between the highly source specific levoglucosan and light absorption measurement derived aethalometer model parameters have so far not been thoroughly investigated. Some studies have found good correlation between levoglucosan and AAE, or calculated BC from wood burning ($BC_{WB}$), using the aethalometer

model (Fuller et al., 2014;Herich et al., 2011;Lack et al., 2013). On the other hand, recently published studies claim that the aerosol spectral dependence is more affected by combustion conditions than the type of fuel being combusted (Garg et al., 2016;Martinsson et al., 2015). Garg et al. (2016) found that the gaseous tracer for biomass burning, acetonitrile, correlated well with AAE during smoldering combustion but poorly during flaming combustion, and further that AAE varied greatly throughout combustion of the same fuel type. Calvo et al. (2015) measured levoglucosan in a wood stove with controlled

combustion and a traditional fireplace; they found elevated concentrations of levoglucosan during the fuel addition followed by a rapid decrease in concentration in the flaming phase. Hence, it is possible that observed correlations between AAE and levoglucosan may only be valid for the smoldering combustion, which may limit the use of both levoglucosan and AAE as universal tracers of biomass burning.

The aethalometer model has so far mainly been applied during winter in highly polluted urban environments (Favez et al.,

2009;Favez et al., 2010;Fuller et al., 2014;Harrison et al., 2013;Sandradewi et al., 2008a). There is thus a lack of knowledge regarding the performance of the aethalometer model during summer, and in less polluted rural environments. For instance, it is not known how the aethalometer model will cope with the usually dominating and presumably non-light absorbing biogenic secondary organic aerosol (SOA) during summer time.





This study was initiated with the aim to compare a light-absorption source apportionment technique, the aethalometer model (Sandradewi et al., 2008a), to traditional filter-based chemical and physical analysis source apportionment using radiocarbon and levoglucosan measurements for a whole year at a rural measurement station in southern Sweden.

## 2 Methods

### 2.1 Measurement site and sampling

Sampling of atmospheric aerosols was conducted at the aerosols, clouds and trace gases research infrastructure (ACTRIS) and European monitoring and evaluation programme (EMEP) rural background station Vavihill, located in southern Sweden (56°01' N, 13°09' E, 172 meters above sea level). The surrounding landscape consists of coniferous and deciduous forests, farmland and pastures. The measurement station is placed on a pasture that is visited by grazing cattle during spring, summer and fall. The closest large cities are Helsingborg, Malmö and Copenhagen which are located at distances of 20, 50 and 65 km in the west to southwest direction, respectively. Aerosols were sampled with a $PM_{10}$-inlet on pre-heated (900 °C for 4 h in air) 47 mm quartz filters (Pallflex 2500QAT-UP) using a sampling time of 72 h at a flow rate of 38 liters per minute (lpm) with an automatic Leckel SEQ47/50 sampler. Active carbon denuders were installed in the sampling line with the purpose of removing volatile organic compounds (VOCs). After sampling, filters were put in petri dishes, wrapped in aluminium foil and stored in a freezer at -18 °C until analysis. The total measurement period lasted from June 2014 until June 2015 and included in total 123 filter samples. The measurement period was divided into seasons with a 3 months intervals, summer=June-August, fall=September-November, winter=December-February and spring=March-May.

### 2.2 OC/EC analysis

Elemental carbon (EC), organic carbon (OC) and total carbon (TC) were measured through thermal-optical analysis (TOA) with a DRI Carbon analyzer (Model 2001). The EUSAAR_2 analytic protocol was used for the analysis (Cavalli et al., 2010). In short, OC from a 0.5 $cm^2$ filter punch is evolved in four different temperature steps in an inert helium atmosphere at a maximum temperature of 570 °C. A 633 nm He/Ne-laser is irradiating the filter and the light transmission through the filter is measured during the increase of temperature. When the measured light transmission reaches its initial base-line value the remaining carbon is considered to be EC. EC is evolved in an oxidizing atmosphere (2 % $O_2$) during high temperatures (500-850 °C). All carbon is oxidized and evolved from the filter as $CO_2$, which is further converted to methane and finally quantified with a flame ionization detector (FID). TC is the sum of OC and EC. Cavalli et al. (2016) estimated the combined random uncertainties from inter-laboratory comparisons between 2008-2011 to be 17 % relative standard deviations (RSD) for Vavihill TC measurements.





### 2.3 Light absorption measurements and the aethalometer model

Aerosol light absorption was measured with an aethalometer (AE33, Magee Scientific) (Drinovec et al., 2015). The aethalometer utilizes an airflow through a filter where particles are deposited. The filter deposition spot is irradiated with seven LEDs of different wavelengths (370, 470, 520, 590, 660, 880, 950 nm) and the attenuation is calculated per unit of time. In this campaign the aethalometer was operating with a flow of 5 liters per minute through a $PM_{10}$ inlet at a time resolution of 1 min. Two main measurement artefacts are associated with filter-based light absorption techniques; the shadowing effect and the filter matrix scattering effect (Weingartner et al., 2003). The AE33 aethalometer handles these artefacts in two ways: attenuation enhancement due to filter matrix scattering is compensated by a factor 1.57, and the shadowing effect is treated by measuring the attenuation at two filter deposition spots with different depositions rates (Drinovec et al., 2015).

The output data of the aethalometer are absorption coefficients, $b_{abs}(\lambda)$, in the units of $m^{-1}$. $b_{abs}(\lambda)$ can be converted into BC mass concentration units ($g\ m^{-3}$) by division of the mass absorption coefficient (MAC), $\sigma_{abs}(\lambda)$ ($m^2\ g^{-1}$) according to Eq (1):

$$BC(\lambda) = \frac{b_{abs}(\lambda)}{\sigma_{abs}(\lambda)} \tag{1}$$

In the aethalometer model (Sandradewi et al., 2008a), the entire aerosol light absorption is assumed to come from fossil fuel combustion aerosol (FF) or wood burning aerosol (WB):

$$b_{abs}(\lambda) = b_{absFF}(\lambda) + b_{absWB}(\lambda) \tag{2}$$

$$\frac{b_{absFF}(370\text{nm})}{b_{absFF}(950\text{nm})} = \left(\frac{370}{950}\right)^{-AAE_{FF}} \tag{3}$$

$$\frac{b_{absWB}(370\text{nm})}{b_{absWB}(950\text{nm})} = \left(\frac{370}{950}\right)^{-AAE_{WB}} \tag{4}$$

By combining Eq. (2-4), it is now possible to calculate the light absorption that is caused by WB and FF in 370 and 950 nm (Mohr et al., 2013;Zotter et al., 2016), respectively:

$$b_{absFF}(950nm) = \frac{b_{abs(370nm)} - b_{abs(950nm)} \cdot \left(\frac{370}{950}\right)^{-AAE_{WB}}}{\left(\frac{370}{950}\right)^{-AAE_{FF}} - \left(\frac{370}{950}\right)^{-AAE_{WB}}} \tag{5}$$

$$b_{absWB}(370nm) = \frac{b_{abs(370nm)} - \left(\frac{370}{950}\right)^{-AAE_{FF}} \cdot b_{abs(950nm)}}{1 - \frac{\left(\frac{370}{950}\right)^{-AAE_{FF}}}{\left(\frac{370}{950}\right)^{-AAE_{WB}}}} \tag{6}$$

In Eq. (3-6) the AAE is the source specific spectral dependence. In the aethalometer model, the selection of source specific AAEs ($AAE_{FF}$ and $AAE_{WB}$) are crucial for accurate source contribution estimation. Traditionally, is has been assumed that pure black carbon is dominating fossil fuel combustion emission, leading to an $AAE_{FF}=1$. Wood burning emissions have previously been assumed to have an AAE around 2 (Kirchstetter et al., 2004). However, recent studies have shown that it is the combustion conditions rather than the fuel itself that determines the organic content in the aerosol, and consequently the AAE (Garg et al., 2016;Martinsson et al., 2015). Martinsson et al. (2015) found that flaming combustion in a modern conventional wood stove emitted aerosol with highly agglomerated soot structure and an AAE of 1.3. Garg et al. (2016)



determined the combustion efficiency by analysing emission gas data and reached similar conclusions. We estimated $AAE_{FF}$ and $AAE_{WB}$ based on literature data (Table 1). From Table 1 a mean $AAE_{FF}=1.0$ (SD=0.1) and mean $AAE_{WB}=1.81$ (SD=0.52) was chosen in this study. The value of $AAE_{WB}=1.81$ is close to the values chosen by Massabo et al. (2015) and Sandradewi et al. (2008a), i.e. 1.81 and 1.86, respectively.

5 By using Eq. (5-6) it is possible to calculate the light absorption due to FF ($b_{absFF}(\lambda)$) or WB ($b_{absWB}(\lambda)$). These light absorption coefficients can then be divided with the site specific MAC (Table 2) in order to calculate the BC mass concentration from each source (Eq. 7-8):

$$BC_{FF} = \frac{b_{absFF}(950nm)}{\sigma_{abs}(950nm)} \tag{7}$$

$$BC_{WB} = \frac{b_{absWB}(370nm)}{\sigma_{abs}(370nm)} \tag{8}$$

10 $\sigma_{abs}(\lambda)$ is in this case the site-specific mass absorption coefficients for the respective wavelengths which can be found in Table 2. Site specific $\sigma_{abs}(\lambda)$ was determined by linear regression of $b_{abs}(\lambda)$ against elemental carbon (EC) concentration in $PM_{10}$. It is also possible to calculate the carbonaceous aerosol mass (CM) from FF, WB and non-light absorbing secondary organic aerosol. The latter is presumably mostly derived from biogenic sources, hence the acronym $CM_{Bio}$:

$$TC = CM_{FF} + CM_{WB} + CM_{Bio} = C_1 \cdot b_{absFF}(950nm) + C_2 \cdot b_{absWB}(370nm) + CM_{Bio} \tag{9}$$

15 In Eq. (9), $C_1$ and $C_2$ are the slopes from the linear regression of measured total carbonaceous matter (TC) and the light absorption due to FF ($b_{absFF}(950nm)$) and WB ($b_{absWB}(370nm)$), respectively. Previous work has set $CM_{Bio}$ as the intercept when solving the multilinear equation, however this is highly unrealistic since biogenic primary and secondary aerosol formation is seasonal dependent and should vary accordingly (Guenther et al., 1995). We propose an alternative method where $CM_{Bio}$ is allowed to vary outside the suggested regressions (Eq. 10-12). If Eq. (9) is rewritten, a linear regression can 20 be used in order to calculate $C_1$ and $C_2$:

$$\frac{TC}{b_{absWB}(370nm)} = C_1 \cdot \frac{b_{absFF}(950nm)}{b_{absWB}(370nm)} + C_2 + \frac{CM_{Bio}}{b_{absWB}(370nm)} \tag{10}$$

$$\frac{TC}{b_{absFF}(950nm)} = C_2 \cdot \frac{b_{absWB}(370nm)}{b_{absFF}(950nm)} + C_1 + \frac{CM_{Bio}}{b_{absFF}(950nm)} \tag{11}$$

For Eq. (10), $C_1$ can be calculated as the slope of the regression line by setting $TC/b_{absWB}(370nm)$ as the dependent variable and $b_{absFF}(950nm)/b_{absWB}(370nm)$ as the independent. Similar approach can be applied to Eq. (11) to calculate $C_2$. By 25 selecting only winter data for calculation of $C_1$ and $C_2$ the interference of $CM_{Bio}$ is minimized and the division of $CM_{Bio}$ by one of the light absorption parameters is forcing $CM_{Bio}$ towards zero. Hence, the intercept of the linear regression line should be close to $C_2$ when calculating the slope as $C_1$ in Eq. 10, and vice versa for Eq. 11. The linear fits used to derive $C_1$ and $C_2$ contained one suspected outlier each. These outliers were confirmed by Grubbs test (Grubbs, 1950) for both dependent and independent variables with 95% confidence and hence removed. The linear fits without outliers are displayed in Fig. S1 and 30 S2. Finally, $CM_{Bio}$ is allowed to vary outside the linear regressions:

$$CM_{Bio} = TC - CM_{FF} - CM_{WB} \tag{12}$$



Since $CM_{Bio}$ is assumed to be the residual carbonaceous matter, i.e. the carbonaceous matter that does not absorb light, this parameter may have a negative value during winter when the sum of $CM_{FF}$ and $CM_{WB}$ exceeds TC.

$C_1$ was calculated to be 214 467 µg m$^{-2}$ with an intercept of 133 794. $C_2$ was estimated to 113 881 µg m$^{-2}$ with an intercept of 273 603. Hence, $C_1$ (from Eq. 10) was deviating with 22 % from the intercept in the calculation of $C_2$ (Eq. 11), while $C_2$ (from Eq. 11) was deviating with 15 % from the intercept in the calculation of $C_1$ (Eq. 10).

Herich et al. (2011) found high standard errors in their modelled $C_1$ and $C_2$ parameters (± 30 %). This was the main reason for Herich et al. (2011) to exclude the CM-approach and proceed with the BC approach presented in Eq. (7-8). In comparison to Herich et al. (2011), we found similar standard error for $C_1$ (31 %) but lower for $C_2$ (18 %). We have therefore chosen to proceed with the CM-approach.

## 2.4 Levoglucosan analysis

1,6-Anhydro-beta-D-glucose (levoglucosan) analysis was performed using the method of Wu et al. (2008) with some modifications. Levoglucosan was purchased from Sigma Aldrich (St. Louise, USA). Hexane from Scharlau (Spain), 1-phenydodecane, 97 % from Acros Organics (Geel, Belgium) and N,O-bis(trimethylsilyl)trifluoroacetamide (BSTFA) containing 1 % trimethylsilyl chrloride (TMCl) was purchased from Sigma (St. Louise, USA). Filter punches were divided into small pieces using a surgical blade and placed in a 50 ml conical flask. Extraction was carried out by sonication using three aliquots of 15 ml, 10 ml and 10 ml of dicloromethane and methanol (1:3) for 45 minutes, 30 minutes and 15 minutes respectively. Extract from each step was filtered and pooled together in a 50 ml beaker using a 0.45 µm polypropylene membrane syringe filter. The total extract was concentrated to dryness under a gentle stream of nitrogen at 60 °C. The final volume of the extract was made up to 1 ml with dicloromethane.

50 µl of each extract was placed in gas chromatography (GC) vials with 300 µl glass inserts and evaporated to dryness under a gentle stream of nitrogen at 60 °C. 15 µl of 1-phenydodecane (97 % Acros Organics, internal standard) solution prepared in hexane (1 µg ml$^{-1}$) and 10 µl of N,O-bis(trimethylsilyl)trifluoroacetamide (BSTFA) containing 1 % trimethylsilyl chrloride (TMCl) were added to the vials (Sigma-Aldrich). The vials were sealed using screw caps with Teflon septa. Samples were derivatized in an oven at 80 °C for 1 h. Solvent blanks and calibration curve were run for each batch of eight samples. Samples were analyzed immediately after derivatization.

An Agilent 6890 series GC with 5973 MS (Agilent Technologies, Palo Alto, USA) was used for the analysis. An Agilent HP-5ms column (30 m x 0.25 mm x 0.25 µm film thickness) was used. Injection volume was 2 µl, splitless, with injector temperature of 280 °C. The temperature program was as follows; initial temperature 60 °C for 3 minutes then the temperature was raised to 190 °C at a rate of 10 °C min$^{-1}$ and then it was finally raised to 300 °C at a rate of 30 °C min$^{-1}$. Transfer line, ion source and quadruple temperatures were 280 °C, 250 °C and 180 °C, respectively. The MS was operated in electron ionization mode. Scan mode was used to identify levoglucosan (99 % pure, Sigma-Aldrich) and 1-phenyldodecane with m/z 217 and 246 respectively. The exact masses used for calibration curves and aerosol samples were determined by



SIM mode as m/z 217.3 and 246.3, respectively. The measurement uncertainty in SD of the GC-MS measurements was estimated to be ±1 % of the levoglucosan peak areas.

## 2.5 $^{14}$C analysis

The $^{14}$C/$^{12}$C ratio in the sampled particles was measured with accelerator mass spectrometry (AMS) (Hellborg and Skog, 2008) by using the 250 kV single-stage AMS at Lund University (Skog, 2007;Skog et al., 2010). Prior to the analysis, the carbon in the particle filter sample was transformed to graphite according to the procedure described in Genberg et al. (2010). In brief, a filter sample corresponding to approximately 50 µg carbon was mixed with CuO and combusted in a vacuum. Evolved $CO_2$ was purified cryogenically, mixed with $H_2$ and heated to 600 °C in the presence of an iron catalyst. In the latter reaction the $CO_2$ was reduced into graphite. The results are presented as fraction modern carbon, $F^{14}C$ (Reimer et al., 2004). A $F^{14}C$ value of 1 represents the 1950 concentration of $^{14}$C excluding human influences. The true atmospheric $^{14}$C content has however been altered due to two effects, known as the bomb effect (Rafter and Fergusson, 1957) and the Suess effect (Suess, 1955). The bomb effect, which is referring to atmospheric testing of thermonuclear weapons in the 1940-1960s, has had a positive effect on the $F^{14}C$ values, due to neutron-induced reactions forming $^{14}$C. The Suess effect is the result of emission of $CO_2$ from anthropogenic fossil fuel combustion, leading to the ongoing increase of the atmospheric $CO_2$ concentration. Since fossil fuels are $^{14}$C-free, the Suess effect generates decreased $F^{14}C$ values of atmospheric carbon (Baxter and Walton, 1970). Estimated measurement uncertainties expressed as SD, are typically ±1 % of measurement values.

Prior to the $F^{14}C$ measurements, 104 out of 123 filter samples were pooled with a neighbouring sample due to limited amount of filter material. In the pooled samples, filter material corresponding to 25 µg C were punched out from each of the two filter samples, resulting in the desirable mass of 50 µg C. Two pooled samples (19th-25th December 2014 and 17th-23rd February 2015) were omitted due to failure in the graphitization process and consequently lack of filter material.

Evaluation of the aethalometer model results was performed using mainly $F^{14}C$ data and the source apportionment approach by Bonvalot et al. (2016). The ambient carbonaceous aerosol can be assumed to be composed of one fossil and one non-fossil fraction. Determination of the non-fossil fraction ($f_{NF}$) is performed by normalizing the measured $F^{14}C$ ($F^{14}C_S$) by a non-fossil reference value ($F^{14}C_{NF,ref}$):

$$f_{NF} = \frac{F^{14}C_S}{F^{14}C_{NF,ref}} \tag{13}$$

A previous source apportionment study at Vavihill suggests that winter samples are highly influenced by wood burning but low levels of other modern carbon sources, i.e. biogenic primary and secondary aerosol (Genberg et al., 2011). Biomass used for wood burning has usually had a growth period of decades, implying that the integrated average $F^{14}C$ for wood burning is higher than the atmospheric $F^{14}C$ at the time of sampling. As in previous studies, we also assume that the biomass used in wood burning has an average $F^{14}C_{WB}$ of 1.10 (Bonvalot et al., 2016;Szidat et al., 2006). Hence, we use $F^{14}C_{NF,ref}$=1.10 during winter.





Summer carbonaceous aerosol mass in Scandinavia has been found to be totally dominated by biogenic primary and secondary organic aerosol (Genberg et al., 2011;Yttri et al., 2011a). Hence, the summer time carbonaceous aerosol should have a $F^{14}C$ close to the atmospheric value at the sampling time, i.e. $F^{14}C_{Bio}$=1.04. Thus, summer time $F^{14}C_{NF,ref}$ was set to 1.04. Spring and fall are characterized by highly mixed sources of modern carbon. It can be expected that both wood burning

and biogenic emissions contribute significantly to the carbonaceous mass during these seasons. We therefore chose the mean of winter and summer $F^{14}C_{NF,ref}$ to represent the spring and fall samples, i.e. 1.07. The total carbon can be assumed to be derived from three possible sources:

$$TC = TC_{NF} + TC_{FF} = TC_{WB} + TC_{Bio} + TC_{FF} \tag{14}$$

In Eq (14), sample TC is divided into non-fossil (NF) and fossil fractions (FF). NF can be further subdivided into wood

burning (WB) and biogenic carbon (Bio). From Eq. (14) it is now possible to set up the $^{14}C$ mass balance equation:

$$TC \cdot F^{14}C_S = TC_{WB} \cdot F^{14}C_{WB} + TC_{Bio} \cdot F^{14}C_{Bio} + TC_{FF} \cdot F^{14}C_{FF} \tag{15}$$

In Eq. (15) $F^{14}C_S$ is the sample $F^{14}C$. $F^{14}C_{WB}$, $F^{14}C_{Bio}$ and $F^{14}C_{FF}$ are the reference $F^{14}C$ value for each of the respective sources. Since $F^{14}C_{FF}$ is equal to zero, this gives:

$$TC \cdot F^{14}C_S = TC_{WB} \cdot F^{14}C_{WB} + TC_{Bio} \cdot F^{14}C_{Bio} \tag{16}$$

TC non-fossil (TC$_{NF}$) can be calculated by Eq. (17).

$$TC_{NF} = TC \cdot f_{NF} \tag{17}$$

Total carbon from wood burning (TC$_{WB}$) can then be calculated by Eq. (18).

$$TC_{WB} = a \cdot [levoglucosan] \tag{18}$$

Here, $a$ is the slope from the linear fit between TC$_{NF}$ and levoglucosan for winter samples (Fig. S3), [levoglucosan] is the

sample levoglucosan concentration. Only winter samples are used and the linear fit is forced through origin with the purpose of minimizing the effect of biogenic carbon on TC$_{NF}$. Hence, we assume that all non-fossil carbon is derived from wood burning during winter. However, it should be noted that combustion of fossil lignite (i.e. brown coal), can emit large quantities of levoglucosan and be confused with wood combustion (Fabbri et al., 2008). It is now possible to calculate the total carbon from biogenic sources:

$$TC_{Bio} = \frac{F^{14}C_S \cdot TC - TC_{WB} \cdot F^{14}C_{WB}}{F^{14}C_{Bio}} \tag{19}$$

In Eq. (19), $F^{14}C_S$ is the sample $F^{14}C$, $F^{14}C_{WB}$=1.10 and $F^{14}C_{Bio}$=1.04 (Bonvalot et al., 2016). Finally, it is possible to derive TC$_{FF}$:

$$TC_{FF} = TC - TC_{WB} - TC_{Bio} \tag{20}$$

**2.6 HYSPLIT**

The Hybrid Single Particle Lagrangian Integrated Trajectory Model (HYSPLIT) (Draxier and Hess, 1998;Stein et al., 2015) was used to study the history of the air mass carrying the particles sampled on the filters and measured by the aethalometer. Gridded meteorological data from the Centre of Environmental Predictions (NCEP) Global Data Assimilation System





(GDAS) were used as input to trajectory model. Back-trajectories were calculated at an hourly frequency 120 h backward in time and the trajectories started 100 m above ground at the Vavihill measurement site. For each filter sample, 72 trajectories were used since the sampling time was 72 h. Four regions of origin (Table 3) were defined and for each sample it was investigated how much time the air-mass had spent over each of the regions of origin. Also, the accumulated precipitation along each trajectory during the last 24 h before arrival at Vavihill was used to evaluate the effect of precipitation on aerosol particle concentration. For each sample, the accumulated precipitation along each of the 72 the trajectories were summarized.

## 2.7 Auxiliary measurements

$NO_X$ was continuously monitored with a time resolution of 1 h with a CLD 700 AL chemiluminescence analyzer (Eco Physics, Duernten, Switzerland). The detection limit was 1 ppb. Mass concentration of $PM_{10}$ was monitored with a tapered element oscillating microbalance (TEOM, 8500 FDMS, ThermoFisher Scientific). The time resolution was 1 h and detection limit 0.1 µg m$^{-3}$.

## 3 Results and discussion

### 3.1 Variations and features in carbon concentrations

Carbonaceous aerosol constitutes on average 13 % of the total $PM_{10}$ during the measurement period. Figure 1a-b displays the temporal variation of particulate carbon throughout the measurement period. OC is elevated during summer (mean=1.29 µg m$^{-3}$) and fall (mean=1.86 µg m$^{-3}$) and then decrease during winter (mean=0.96 µg m$^{-3}$) and spring (mean=1.19 µg m$^{-3}$). EC peaked during fall (mean=0.32 µg m$^{-3}$) while the concentrations during winter (mean=0.19 µg m$^{-3}$), spring (mean=0.21 µg m$^{-3}$), and summer (mean=0.14 µg m$^{-3}$) were significantly lower (p<0.05, Fig. 1b).

A discrepancy in winter concentrations of carbonaceous compounds was found between this study (2014-2015) and a previous Vavihill source apportionment study (2008-2009). Genberg et al. (2011) found elevated concentrations of OC during winter (2.19 µg m$^{-3}$, p<0.1) and approximately twice the amount of EC during winter compared to summer (0.30 µg m$^{-3}$ vs. 0.14 µg m$^{-3}$, p<0.001). EC is typically elevated during the cold period of the year, when residents burn wood for heating. In the present study we found no significant differences in OC or EC between summer and winter. Figure 1a reveals a decrease in carbonaceous aerosol mass concentration during winter, from the beginning of December 2014 to mid-February 2015. In fact, by comparing the measurement campaign TC during the winter 2014-2015 with the average TC during earlier winters we found that the TC concentration during the winter 2014-2015 was 35 % lower than the average winter of 2008-2013 (p=0.024, Fig. S4). By using HYSPLIT we find that the incoming air masses to Vavihill during the winter of this measurement campaign were influenced by approximately 45 % more precipitation than the average winter of 2000-2013 (p=0.002, Fig. S5). Furthermore, we found a weak but significant negative relationship between precipitation and TC ($R^2$=0.1; p<0.05). Wet deposition losses are thus likely to be at least a partial explanation of lower winter-time concentrations of carbonaceous aerosol in this study.



### 3.2 Variations in light absorption measurements and aethalometer model derived parameters

Figure 2 shows the AAE throughout the whole measurement campaign (June 2014–June 2015). In general, there was a strong negative relationship between AAE and ambient temperature ($R^2$=0.74; p<0.001). During summer the AAE remains low in the range of 1.0-1.2 (mean=1.12; standard deviation=0.07). An increase in AAE follows during the fall (mean=1.23;

SD=0.1) and stays at 1.2-1.5 (mean=1.36; SD=0.07) throughout the winter period. In the spring, the AAE remains high (mean=1.31; SD=0.09), but is decreasing towards 1.1-1.2 at the end of the season. There is a significant difference in AAE between all seasons (p<0.01), except between winter and spring (p=0.055). The observed seasonal pattern is in accordance with earlier studies by Sandradewi et al. (2008b) and Herich et al. (2011) who found elevated AAE of 1.3-1.6 during winter and a decreased AAE of around 1.0 during summer.

Elevated AAE during the cold period of the year is most likely caused by increased use of wood burning for residential heating, this has been confirmed in several studies (Genberg et al., 2011;Herich et al., 2011;Sandradewi et al., 2008b). Since the measured aerosol light absorption most likely is a mixture of fossil and wood burning the selection of AAEs for the aethalometer model calculations are supported by the observed seasonal pattern, i.e. the observed AAE vary between $AAE_{FF}$ and $AAE_{WB}$.

Figure 3 shows the diurnal variation of AAE (370-950 nm), $BC_{FF}$, $BC_{WB}$ and $NO_X$ between summer and winter and between weekdays and weekends. There is a minimum in AAE at 7-10 AM (local time, Fig. 3a-b), this coincide with morning traffic rush hours. Sandradewi et al. (2008b) found similar results during winter with a minimum in AAE around 8 AM. This result is confirmed by data presented in Fig. 3d, which shows the calculated $BC_{FF}$ (950nm) from the aethalometer model. It is clear that there is a peak at 8-10 AM and at 5-7 PM in the $BC_{FF}$ emissions. This pattern is validated by $NO_X$ concentrations

showing the similar pattern as $BC_{FF}$ (Fig. 3g-h). Rissler et al. (2014) found similar peaks in $NO_X$ and BC from a busy road in Copenhagen. A major source of $NO_X$ is vehicle combustion engines and $NO_X$ can thus be expected to correlate with $BC_{FF}$. Due to the rural location of Vavihill measurement station, it may take 2-3 h for the traffic emissions to reach the station if they originate from the major cities in the region, this can explain why the $NO_X$ (and $BC_{FF}$) peaks occur somewhat later at Vavihill than expected traffic rush hours. Studying the long term pattern between $BC_{FF}$ and $NO_X$, there is a weak but

significant correlation throughout the whole measurement campaign ($R^2$=0.09; p<0.001). However, since $NO_X$ is efficiently oxidized by OH-radicals and ozone during periods with high UV-radiation it is more suitable to compare these on a seasonal basis. Significant but very weak correlations between $BC_{FF}$ and $NO_X$ were found during fall ($R^2$=0.07; p=0.021), winter ($R^2$=0.2; p<0.001), spring ($R^2$=0.41; p<0.001) and summer ($R^2$=0.09; p=0.009). The $CM_{FF}$ parameter shows similar pattern as $BC_{FF}$ to $NO_X$.

The $BC_{WB}$ concentration has a different diurnal pattern than $BC_{FF}$. In general, there is a peak in the $BC_{WB}$ concentration from 7 PM to 3 AM (Fig. 3f), which indicates that most residents warm their houses by wood burning during the evenings and nights. Previous studies have found a similar diurnal pattern for wood burning derived emissions (Favez et al., 2010;Harrison et al., 2012;Harrison et al., 2013;Kristensson et al., 2013;Wang et al., 2011).





$NO_X$ is not thought to be emitted in large quantities from wood burning, still $NO_X$ and $CM_{WB}$ concentrations are correlated during the whole measurement period ($R^2$=0.31; p<0.01). This can be explained by the fact that both parameters are strongly seasonal dependent, but for different reasons, thus correlation is strong but causality is most likely absent between them.

The seasonal patterns of other aethalometer model derived parameters are presented in Table 4 and Fig. 4. It is clear that the wood burning derived carbonaceous aerosol, $CM_{WB}$, follows a seasonal cycle with high concentrations during fall (mean=0.49 µg m$^{-3}$), winter (mean=0.65 µg m$^{-3}$) and spring (mean=0.51 µg m$^{-3}$), and low levels during summer (mean=0.07 µg m$^{-3}$). The $CM_{WB}$ peaks with 5 % contribution to $PM_{10}$ during winter, in summer the $CM_{WB}$ contribution is low (0.6 %). The $CM_{WB}$ contribution to TC peaks in winter with 56 % and is reduced to 6 % during summer. Hence, it is likely that the largest part of wood burning is conducted with the purpose of residential heating, as in contrast to decorative burning which can be expected independently of outdoor temperature.

The fossil fuel derived parameter, $CM_{FF}$, shows a less distinct seasonal pattern than $CM_{WB}$, most probably because the main source, traffic, has a much smaller seasonal variation than wood burning. $CM_{FF}$ contributed with 2-4 % to $PM_{10}$ during the year (21-35 % contribution to TC) with a maximum during spring and a minimum during summer. Finally, the biogenic aerosol carbon concentrations are substantial during summer (9 % of $PM_{10}$; 72 % of TC) and low during winter (0.9 % of $PM_{10}$; 8 % of TC).

### 3.3 Comparison: Levoglucosan to aethalometer model

Levoglucosan concentrations displayed an annual variation with elevated concentrations during the cold period of the year (Fig. S6). Mean concentrations during fall, winter and spring were 0.061 (SD±0.082), 0.086 (SD±0.073) and 0.063 (SD±0.115) µg m$^{-3}$, respectively. The summertime mean levoglucosan concentration was 0.014 (SD±0.0142) µg m$^{-3}$. There was a significant difference between winter and summer (p=0.03). Measured concentration levels and seasonal patterns were similar to those found by Genberg et al. (2011) at the same measurement site. The aethalometer model derived carbonaceous matter from wood burning, $CM_{WB}$, correlated well with levoglucosan (Fig. 5, $R^2$=0.7; p<0.001). The correlation was strongest during winter ($R^2$=0.82) and spring ($R^2$=0.81) and lower during fall ($R^2$=0.37) and summer ($R^2$=0.30). Mean measured levoglucosan per unit of $BC_{WB}$ was estimated to 0.64 (standard deviation=0.73). Previous wood stove measurements report mean levoglucosan to EC ratio of 0.82 (Iinuma et al., 2007;Schmidl et al., 2008). Thus, the estimated ratios presented in this study are in line with emission inventories from wood stoves. The measured ratios in comparison to references imply that the atmospheric decomposition of levoglucosan is in general slow, at least during the cold seasons. Another possibility is that the wood burning sources are located fairly close to the sampling site.

Further, $CM_{FF}$ also correlated with levoglucosan ($R^2$=0.39; p<0.001). This finding is in contrast to Herich et al. (2011) who found no correlation between $BC_{FF}$ and levoglucosan in the alpine regions of Switzerland. One explanation might be inaccurate apportionment where the wood burning aerosol exhibits an AAE close to 1, and thus being apportioned as fossil fuel aerosol. This hypothesis is supported by the study of Martinsson et al. (2015), but whether this phenomenon would be more common and pronounced in Swiss alpine regions in comparison to southern Sweden is unknown.



### 3.4 Aethalometer model evaluation by radiocarbon and levoglucosan source apportionment

We used $F^{14}C$ and levoglucosan data (Fig. S6-7) applied to the method proposed by Bonvalot et al. (2016) to evaluate the aethalometer model parameters. In Eq. (18), $a$ was set to 8.32 based on results from linear regression between winter values of $TC_{NF}$ and levoglucosan. The apportioned fossil fuel carbon fraction from the $F^{14}C$ and levoglucosan method ($TC_{FF}$) is

estimated to 20 % of TC throughout the year (Fig. 6), this is in good agreement with the previous studies from Vavihill measurement station (Genberg et al., 2011;Yttri et al., 2011a). However, there was a significant difference in fossil carbon apportioned between the two methods ($p=0.007$). Throughout the year, the aethalometer model overestimates the fossil carbon by a factor 1.3 compared to $F^{14}C$ and levoglucosan source apportionment. It is possible that this small discrepancy is derived from the relatively low $R^2$ value ($R^2=0.29$) in the fit used to estimate the $C_1$ parameter. Further, $TC_{FF}$ displays a

better agreement with $NO_X$ than the aethalometer model derived $CM_{FF}$ ($R^2=0.15$; $p<0.001$ vs. $R^2=0.06$; $p=0.007$), indicating a more accurate apportionment of fossil carbon using $F^{14}C$.

Apportioned wood burning, $TC_{WB}$ showed, as the aethalometer model $CM_{WB}$, a clear intra-annual variability with high carbon contribution during winter (60 %) and low during summer (9 %). There was no significant difference in apportioned wood burning carbonaceous aerosol between the two methods ($p=0.8$).

The biogenic carbon fraction, $TC_{Bio}$, is dominating TC during summer (75 %), but is not negligible during winter (13 %, Fig. 6) in the radiocarbon and levoglucosan model. Apportioned biogenic carbon was in good agreement between the methods, i.e. no significant differences between the methods were observed for the whole year data ($p=0.32$).

Thus, with respect to apportioned wood burning and biogenic carbonaceous aerosol, the aethalometer model setup presented in this paper shows good agreement with the radiocarbon and levoglucosan model. However, it is interesting to investigate

two other possible setups of the model, for a possibly more accurate aethalometer model source apportionment of the fossil carbon: A) to include the removed outliers in the linear regressions used to derive $C_1$ and $C_2$; B) to solve Eq. 9 with a bilinear fit, as originally proposed by Sandradewi et al. (2008a).

A) Including removed outliers would result in $C_1$ and $C_2$ parameters of 371 047 µg m$^{-2}$ and 88 188 µg m$^{-2}$, respectively. The statistics for both linear regressions would improve, the $R^2$ for $C_1$ would for instance increase from 0.29 to

0.67. However, increasing the $C_1$ parameter by a factor of 1.7 (from 214 467 to 371 047 µg m$^{-2}$) would result in large discrepancies compared to the $F^{14}C$ and levoglucosan method. In general, for the whole measurement campaign, the fossil fuel contribution by the aethalometer model would be overestimated by a factor 2.4 while the biogenic mass contribution would be underestimated by a factor 1.7 compared to radiocarbon and levoglucosan source apportionment. The $CM_{WB}$ contribution to TC would be underestimated by a factor 1.3 compared to $TC_{WB}$.

B) When we derived the $C_1$ and $C_2$ parameters by solving Eq. 9 as a multilinear fit (letting $CM_{Bio}$ be a fixed intercept) $C_1$ and $C_2$ were determined to 497 279 µg m$^{-2}$ and 68 859 µg m$^{-2}$, respectively. $CM_{Bio}$ was fixed to -0.103 µg m$^{-3}$. Hence, the multilinear solution provides a $C_1$ parameter that is approximately 2.2 times larger than the $C_1$ obtained by the current linear regression of Eq. 10, and a $C_2$ parameter that is 1.7 times smaller than the $C_2$ obtained by Eq.



11. The multilinear aethalometer model solution should ideally be compared to radiocarbon and levoglucosan source apportionment results derived from Eq. 18 were *a* was derived from a linear fit of winter data with an allowed intercept, i.e. biogenic carbon is allowed in $TC_{NF}$. *a* is then determined to 7.16 with an intercept of 0.17. This approach will lead to an overestimation of $CM_{FF}$ by a factor 3.2 compared to $TC_{FF}$ and an underestimation of $CM_{WB}$ by a factor 1.5 compared to $TC_{WB}$. Thus, a bilinear solution to Eq. 9 would increase the discrepancy between the aethalometer model and the radiocarbon and levoglucosan source apportionment method.

## 3.5 Air mass trajectory analysis

For the whole measurement period, air masses arriving from SW dominated and contributed with 35 % of the incoming air masses. The remaining contributions of the NW, SE and NE sectors were 32, 17 and 16 %, respectively. Air masses arriving from SE were dominating during the fall (31 %) and were more polluted than air masses from other directions. $CM_{FF}$, $CM_{WB}$, levoglucosan and $PM_{10}$ all increased with increasing fraction of incoming SE winds (p<0.01). The elevated PM levels from this area can be explained by a large fraction of densely populated land and with air dominated by high pressure systems inhibiting vertical mixing with cleaner air. Increasing fraction of NE air masses correlated with increasing amount of biogenic aerosol ($CM_{Bio}$, p<0.01), while other types of PM were low. $F^{14}C$ also increased with NE fraction (p=0.03), supporting the impact of biogenic sources from this direction. This relation seems geographically sound, due to the relatively large and sparsely populated land area. The NW direction from Vavihill is dominated by the North Atlantic, North Sea and Norwegian Sea which are displayed in the results; all carbonaceous PM species tend to decrease with increasing fraction of incoming air mass from NW (p<0.01). Air masses arriving from this region can thus be considered relatively clean, this direction dominated during the summer (43 %). Finally, SW air masses tend to carry $NO_X$, but no carbonaceous PM species correlates with this air mass direction. In general, these results are in line with the findings of Kristensson et al. (2008) who found that air masses from north were in general cleaner than air masses from continental Europe.

## 3.6 Source apportionment uncertainty estimation by the propagation of errors

Many source apportionment studies omit comprehensive uncertainty estimations. This can have severe impacts for decision and policymaking based on the studies. In the present study, an attempt to approximate measurement and linear estimation uncertainty on the calculated fractions of fossil fuel ($CM_{FF}$), wood burning ($CM_{WB}$) and biogenic ($CM_{Bio}$) carbonaceous aerosol was conducted by the recommendations of Henry et al. (1984). The approach of propagation of errors was used and the most obvious uncertainties were estimated and summarized. Aethalometer measurements have been suggested to give an uncertainty of 5 % to absorption coefficients (Hansen, 2005). However, recent work by Zanatta et al. (2016) proposes an uncertainty of 35 % to aethalometer derived absorption coefficients. We select the more conservative uncertainty of 35 % for this analysis. $AAE_{WB}$ and $AAE_{FF}$ are associated with uncertainties of 30 % and 10 %, respectively (Table 1). It should be noticed that previous studies support our selection of AAEs (Massabo et al., 2015;Sandradewi et al., 2008b). The measured values of OC, EC and TC are associated with an uncertainty of 17 % (Cavalli et al., 2016). Finally, the estimation of the $C_1$





and $C_2$ parameters were associated with uncertainties of 31 % and 18 %, respectively. Considering that the fraction of fossil fuel combustion aerosol is based on aethalometer measurements (absorption coefficients), $AAE_{FF}$, TOA (OC, EC, TC) and $C_1$, this parameter get a total uncertainty of 41 %. Similarly, for the fraction wood burning aerosol, we base the total uncertainty on aethalometer measurements, $AAE_{WB}$, TOA and $C_2$. The overall uncertainty is then estimated to 42 %. Despite

the relatively high estimated uncertainty, it is worth noting that the $CM_{WB}$ agreement with levoglucosan was satisfactory (Fig. 5). Finally, we base the biogenic aerosol ($CM_{Bio}$) uncertainty on aethalometer measurements, $AAE_{FF}$, $AAE_{WB}$, $C_1$, $C_2$ and TOA. The biogenic carbonaceous aerosol fraction then reaches a total uncertainty of 50 %.

## 4 Conclusions

The aethalometer model offers fast, inexpensive apportionment of the carbonaceous aerosol. The accuracy and robustness of

the model principle has previously been questioned. In this study we propose a small modification to the aethalometer model which enables apportioned non-absorbing carbon, here assumed to be biogenic carbon, to vary. Propagation of errors showed that fossil, wood burning and biogenic carbonaceous aerosol quantification by the aethalometer model may be highly uncertain. Nevertheless, we show that the model works well for a whole year source apportionment for quantifying wood burning and variable biogenic carbonaceous aerosol at a rural site in southern Sweden, as there was a good agreement

between aethalometer model and the radiocarbon and levoglucosan source apportionment. The aethalometer model overestimated the fossil carbonaceous aerosol compared to the radiocarbon and levoglucosan method, which may be explained by the relatively low $R^2$ value of the linear regression used to relate fossil carbonaceous aerosol light absorption to carbonaceous aerosol mass concentration. However, relating aerosol light absorption to carbon mass concentration by a bilinear solution or including statistically determined outliers resulted in even larger discrepancies between the two methods.

Future studies are needed to investigate the repeatability of our proposed modified aethalometer model.

## 5 Data availability

All data are accessible through the supporting information.

## Author contributions

Johan Martinsson designed the study and analysed all data. Hafiz Abdul Azeem conducted levoglucosan analysis. Moa

Sporre generated the HYSPLIT results. Erik Ahlberg and Emilie Öström were involved in the aerosol sampling. Adam Kristensson, Erik Swietlicki, Kristina Eriksson Stenström and Robert Bergström assisted in the writing process.



## Acknowledgements

This work was supported by The Swedish Research Council FORMAS (project 2011-743). The authors acknowledge research technician Mattias Olsson for graphitization and analysis of radiocarbon in filter samples.

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



**Table 1: Values of AAE$_{FF}$ and AAE$_{WB}$ derived from reference emission inventories.**

| AAE$_{FF}$ | Spectral region | AAE$_{WB}$ | Spectral region | References |
|---|---|---|---|---|
| 1 | 470-660nm | 2.1 | 470-660nm | (Clarke et al., 2007) |
| | | 1.45 | UV-IR | (Day et al., 2006) |
| | | 1.4 | UV-IR | (Garg et al., 2016) |
| 0.9 | UV-IR | 2.5 | UV-IR | (Kirchstetter et al., 2004) |
| | | 1.9 | UV-IR | (Kirchstetter and Thatcher, 2012) |
| | | 1.75 | UV-IR | (Lewis et al., 2008) |
| | | 1.3 | UV-IR | (Martinsson et al., 2015) |
| | | 1.3 | 467-660nm | (Roden et al., 2006) |
| | | 1.6 | UV-IR | (Saleh et al., 2013) |
| | | 2.8 | UV-IR | (Sandradewi et al., 2008b) |
| 1.1 | 450-700nm | | | (Schnaiter et al., 2003) |



**Table 2: Site specific mass absorption coefficients (MAC) from the Vavihill measurement station. Values were obtained by linear regression analysis of measured absorption coefficients and elemental carbon (EC) concentrations. The slope is equivalent to $\sigma_{abs}(\lambda)$. Uncertainties are represented by standard errors (N=123).**

| $\lambda$ (nm) | $\sigma_{abs}(\lambda)$ (m$^2$ g$^{-1}$) |
|---|---|
| 370 | $41.21 \pm 1.00$ |
| 470 | $29.06 \pm 0.96$ |
| 520 | $24.78 \pm 0.80$ |
| 590 | $21.29 \pm 0.68$ |
| 660 | $17.57 \pm 0.55$ |
| 880 | $12.64 \pm 0.39$ |
| 950 | $11.93 \pm 0.37$ |





**Table 3: Definition of wind directions of incoming air masses.**

| Direction | Degrees (°) |
|---|---|
| Northeast (NE) | 0-90 |
| Southeast (SE) | 90-180 |
| Southwest (SW) | 180-270 |
| Northwest (NW) | 270-360 |



**Table 4: Seasonal mean concentrations and contributions to PM$_{10}$ and TC of aethalometer model derived parameters. Uncertainties are given in standard deviations.**

| Season | Concentration (µg m$^{-3}$) | | | Contribution to PM$_{10}$ (%) | | | Contribution to TC (%) | | |
|---|---|---|---|---|---|---|---|---|---|
| | CM$_{WB}$ | CM$_{FF}$ | CM$_{Bio}$ | CM$_{WB}$ | CM$_{FF}$ | CM$_{Bio}$ | CM$_{WB}$ | CM$_{FF}$ | CM$_{Bio}$ |
| **Summer** | 0.07±0.05 | 0.31±0.19 | 1.04±0.59 | 0.6±0.4 | 2.7±1.1 | 9.0±2.8 | 6.1±4.0 | 21.5±6.9 | 72.4±6.5 |
| **Fall** | 0.49±0.46 | 0.62±0.36 | 1.06±0.68 | 2.6±1.9 | 3.7±1.1 | 6.8±3.4 | 22.2±14.9 | 28.4±8.0 | 49.3±19.8 |
| **Winter** | 0.65±0.53 | 0.37±0.25 | 0.13±0.21 | 5.4±3.5 | 3.1±1.3 | 0.9±1.6 | 56.5±13.3 | 35.5±9.4 | 8.0±14.4 |
| **Spring** | 0.51±0.69 | 0.35±0.30 | 0.54±0.32 | 4.9±3.5 | 3.9±2.1 | 8.9±9.6 | 32.3±17.9 | 25.6±9.0 | 42.1±22.5 |





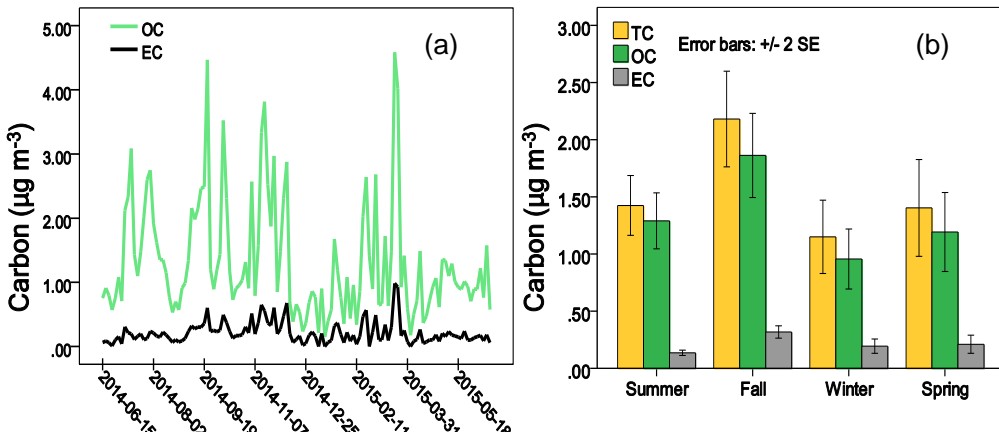

**Figure 1: Temporal variations in OC, EC and TC. (a) Shows the temporal variation of OC and EC with a time resolution of 72 h (N=123). (b) Displays the average concentration of TC, OC and EC divided into seasons; summer (N=32), fall (N=30), winter (N=30) and spring (N=31). Error bars display ±2 standard errors (SE).**



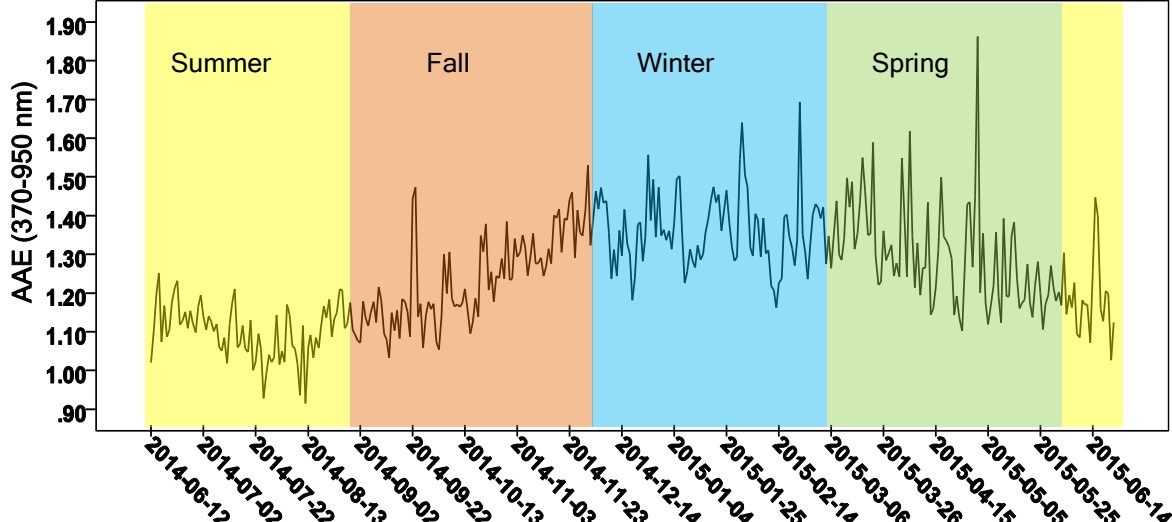

**Figure 2: Annual variations in AAE (370-950nm) at the Vavihill measurement station. Colours represent different seasons of the year. N=369.**





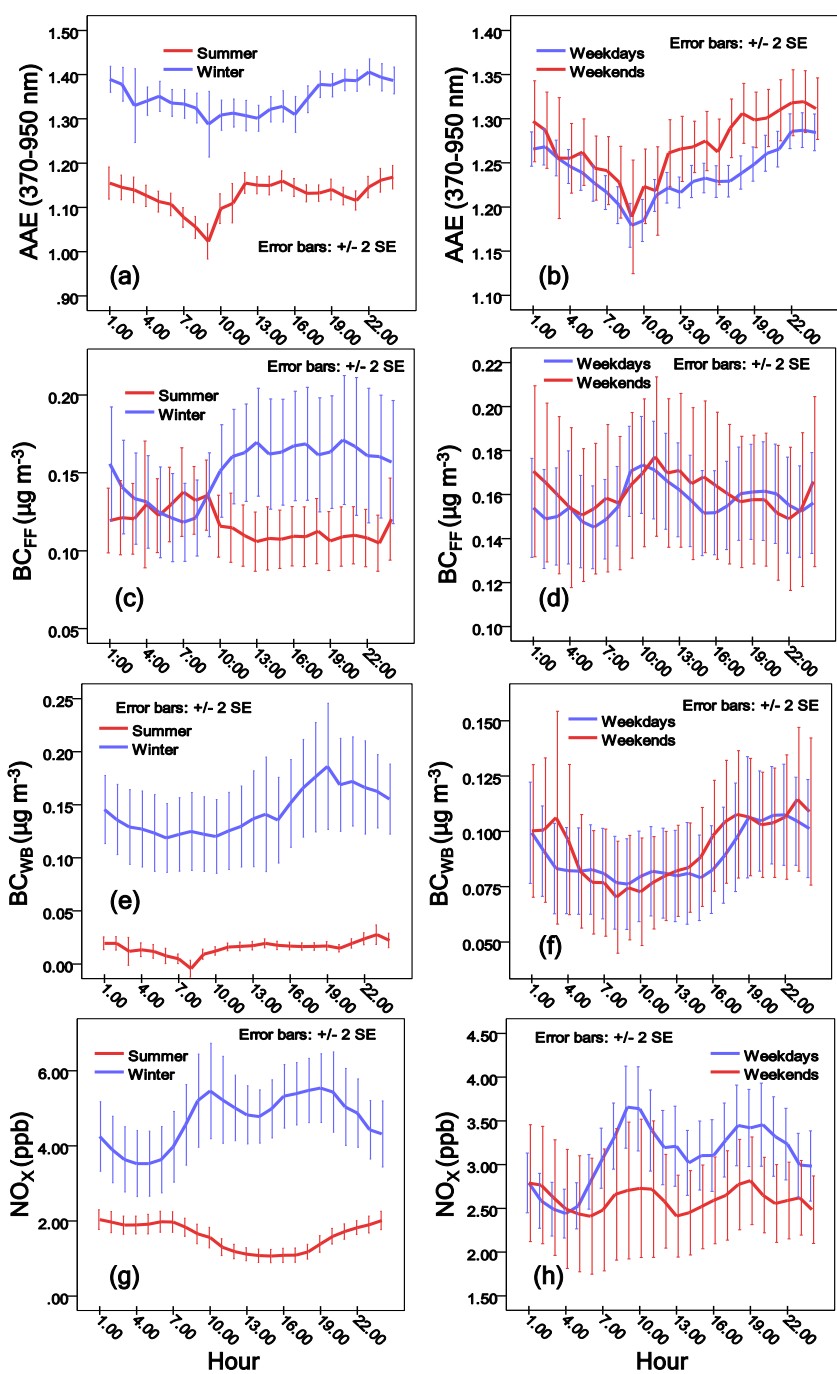

**Figure 3: Diurnal variations of AAE (370-950 nm, a-b), $BC_{FF}$ (950 nm, c-d), $BC_{WB}$ (370 nm, e-f) and $NO_X$ (g-h) at the Vavihill measurement station. Figures (a), (c), (e) and (g) represents diurnal differences between summer and winter while Fig. (b), (d), (f), and (h) represents diurnal differences between weekdays (Monday-Friday) and weekends (Saturday-Sunday, including national holidays). Uncertainties are given as 2 times the standard error (SE).**





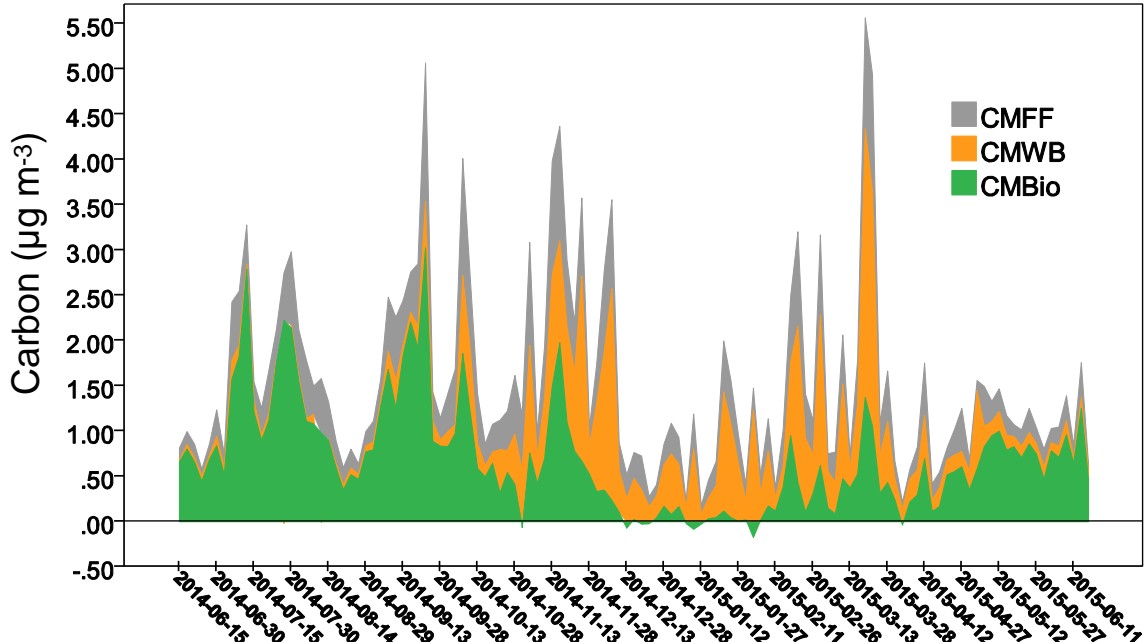

**Figure 4: Aethalometer model source apportionment of total carbon from the Vavihill measurement station, June 2014–June 2015. N=123.**




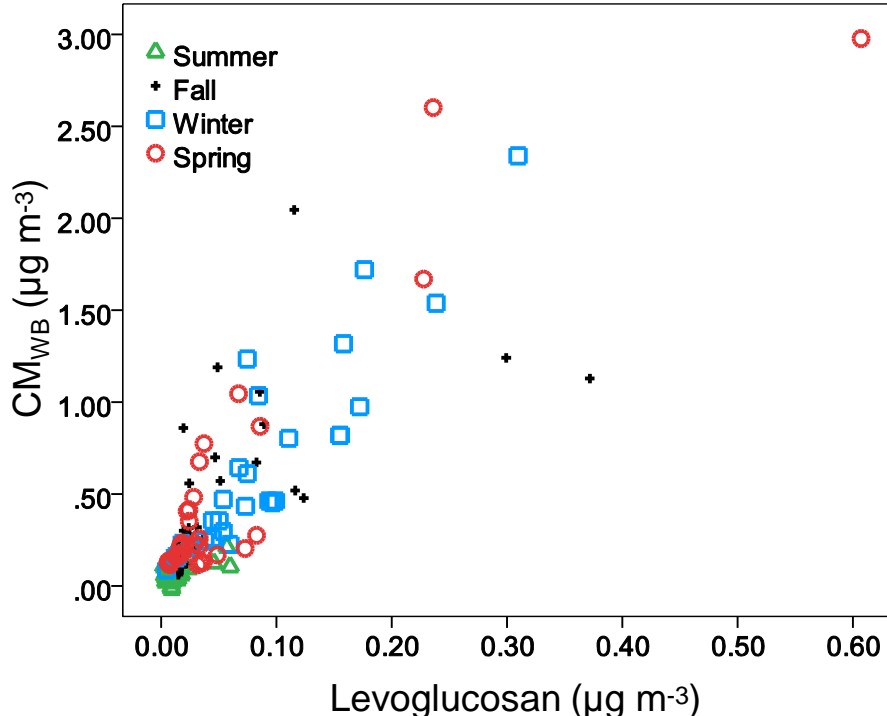

**Figure 5: Comparison between CM$_{WB}$ and levoglucosan. R$^2$=0.70, N=122.**





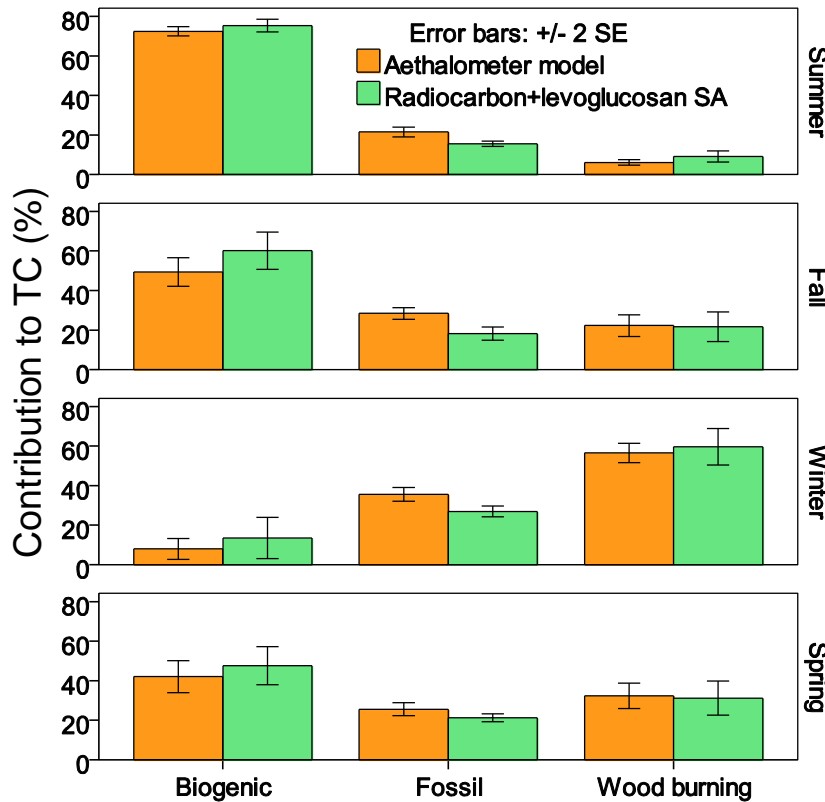

**Figure 6: Seasonal comparison of source contribution to total carbon (TC) between the aethalometer model and radiocarbon + levoglucosan source apportionment (SA). Error bars display 2 times standard error of the mean (SE).**