# Peer review of "Carbonaceous aerosol source apportionment using the aethalometer model-Evaluation by radiocarbon and levoglucosan analysis at a rural background site in southern Sweden"

_Atmospheric Chemistry and Physics, 2016_

## Referee Comment (RC1) · Anonymous Referee #1 · 19 Jan 2017

Martinsson et al. present analysis of a year long dataset of carbonaceous aerosols at a rural background site. Long-term source apportionment studies of carbonaceous aerosols are rare, and in addition the authors propose a new modification to the 'aethalometer model' that they suggest can apportion biogenic sources in addition to wood burning and fossil fuel from aethalometer data. However, while the source apportionment results are promising for wood burning and biogenic carbon, there is a clear shortcoming in its ability to apportion carbon from fossil fuel. The authors are upfront with this issue and have tried alternative methods to improve the source apportionment model, but these did not offer any improvements. I would have liked to see more dis-

cussion on why the proposed aethalometer model over-estimated fossil fuel carbon, apart from the blaming the poor correlation between fossil fuel aerosol light absorption and carbon mass concentration. Why do the authors think there was such a poor fit?

Overall, the paper is well written and structured with a wide range of relevant references and I would recommend it for publication after consideration to the comments below.

Comments:

1. Page 11, line 15-20: While I would agree that the in Fig 3d, the BCFF diurnal trend is bimodal like traffic emissions (though with a peak at night?) I am not sure I agree that the diurnal trend in BCFF for winter especially (Fig 3c), is similar to NOx (Fig 3g). The diurnal trends in NOx in winter are more what would be expected for traffic emissions, and are dissimilar to that observed for BCFF. The flat diurnal trend in BCFF for winter instead to me suggests that the model did not apportion the FF fraction correctly.

2. Page 11, line 25: If NOx is being oxidized in transport to the site, then shouldn't the CMFF and NOx show the highest correlation during winter when there is less photo-chemistry compared to the other seasons? Why is there such a better correlation in spring compared to the other seasons?

3. Page 12, lines 1-3: I find it surprising that the CMwb was better correlated with NOx than CMFF. You explain this by stating that both NOx and CMwb have a seasonal dependence, but why would NOx and CMwb have the same seasonal dependence? Have you examined the correlation between CMwb and NOx to see if there is a change for the seasons?

4. Page 12, line 17. In Fig S6, why there is a very large spike in the concentration of levoglucosan in late March that is not observed in AAE (fig 2)? This sharp spike in levoglucosan suggests a biomass burning event, and I would have thought you would observe a corresponding increase in AAE if wood burning aerosols have a high AAE? Investigating the cause of the spike may help guide the choice of AAE for wood burning

at the site.

5. Section 3.5: did you use annual means for your comparison of the aethalometer and radiocarbon and levoglucosan apportionment? Did you see any changes in agreement between the two methods for the seasons, e.g. was there better agreement in summer or winter?

6. Page13, line 15: In the aethalometer model, to calculate C1 and C2 only winter data was used as it was assumed that there would be negligible CMbio. However, the results from the radiocarbon and levoglucosan model suggest that biogenic carbon was not negligible during winter. I think that you should therefore include some discussion on how the presence of biogenic carbon in winter affected the source apportionment by the aethaolometer model.

7. Page 13, line 31: Why did you fix CMbio to -0.103 ug m-3 and not zero as you expect no CMBIO in winter?

8. Section 3.6: My understanding is that in the proposed aethalometer model, the influence of biogenic carbon needs to be minimized in order to calculate C1 and C2. In addition to only using winter data, could you not also select data for the calculation by wind direction? My reading of this section is that there are geographically distinct areas around the sampling site, and that filtering by wind direction you could further decrease in the biogenic influence (e.g. removing data when the wind is from a forested area or from the NE?) in the data.

9. Page 14, line 19-20: Why would SW air masses have high NOx but not be associated with carbonaceous aerosols, when traffic emissions are a significant source of both?
* * *

---

## Referee Comment (RC2) · Anonymous Referee #2 · 13 Feb 2017

General comments

The authors present an evaluation of the aethalometer model for carbonaceous particle source apportionment using radiocarbon and levoglucosan measurements, and quantify wood burning (WB) and fossil fuel combustion (FF) aerosol for a year-long dataset from a rural station in southern Sweden. The model is modified to allow for apportioned non-light absorbing biogenic aerosol to vary in time, improving the aethalometer-based source apportionment compared to radiocarbon and levoglucosan data. This is an interesting and solid study. The manuscript is very well written (with a few grammatical

errors here and there, I suggest having it checked by a native English speaker), and the analysis is sound. I therefore recommend publication in ACP after the following comments have been addressed:

Specific comments

P. 4, l. 13: How efficient are the active carbon denuders? Please add information on that and the expected background. Is there any information on the evaporation of semi-volatiles from the particles after disturbance of the gas-particle equilibrium due to the denuders?

P. 6, l. 1 – 4: Why were the AAE values not calculated based on the actual data, or at least calculated and compared to literature data? Assuming an AAEFF of 1, and plotting/fitting babs vs wavelength (either averaged, or time-dependent, more appropriate here) can be used to derive AAEWB.

P. 6, l. 11-12: If site specific sigma_abs were calculated based on linear regression of babs against EC, doesn't that imply an overestimation of sigma_abs, as light-absorbing OC is not included in EC? Please clarify.

P. 6, l.13: CM could also be SOA from WB and FF; should be mentioned here.

P. 12, l. 2-3: This statement should be supported by references and more explanations.

P. 12, l. 29: Is the year-long time series correlated, or the diel evolution? I am assuming you are talking about the correlation of the time series. Apart from the non-optimal apportionment, a reason could also be, similar to the correlation of NOx and CMwb, a similar trend, but a different cause. CMFF is higher in winter than in summer, and so is NOx – potentially traffic emissions become more important in winter as well, or meteorological conditions favor the build-up of pollution episodes?

P. 13, l. 15: CMbio was assumed to be 0 for calculations of C1 and C2. Could this be a reason for the overestimation of CMFF?

Technical comments

P. 7, l. 14: typo in chloride (also in subsequent use of trimethylsilyl chloride) P. 7, l.16: typo, should be dichloromethane

---

## Author Comment (AC1) · 9 Mar 2017

Martinsson et al. present analysis of a yearlong dataset of carbonaceous aerosols at a rural background site. Long-term source apportionment studies of carbonaceous aerosols are rare, and in addition the authors propose a new modification to the 'aethalometer model' that they suggest can apportion biogenic sources in addition to wood burning and fossil fuel from aethalometer data. However, while the source apportionment results are promising for wood burning and biogenic carbon, there is a clear shortcoming in its ability to apportion carbon from fossil fuel. The authors are upfront with this issue and have tried alternative methods to improve the source apportionment model, but these did not offer any improvements. I would have liked to see more discussion on why the proposed aethalometer model over-estimated fossil fuel carbon, apart from the blaming the poor correlation between fossil fuel aerosol light absorption and carbon mass concentration. Why do the authors think there was such a poor fit?

We have performed a sensitivity analysis where we increased or decreased the carbon in Eq. 10-11, while not changing the $b_{abs}$ parameters. This would be analogous to change the mass of non-light absorbing carbon. We have written a detailed discussion regarding this in section 3.4. Our results from this analysis suggest that interference of non-light absorbing carbon (presumably biogenic carbon) may be responsible for the observed overestimation of $CM_{FF}$. Hence, we have re-written our abstract, discussion and conclusion where we have added this information. We have omitted that the poor fit would be the reason for the overestimation.

Overall, the paper is well written and structured with a wide range of relevant references and I would recommend it for publication after consideration to the comments below.

Comments:

1. Page 11, line 15-20: While I would agree that the in Fig 3d, the BCFF diurnal trend is bimodal like traffic emissions (though with a peak at night?) I am not sure I agree that the diurnal trend in BCFF for winter especially (Fig 3c), is similar to NOx (Fig 3g). The diurnal trends in NOx in winter are more what would be expected for traffic emissions, and are dissimilar to that observed for BCFF. The flat diurnal trend in BCFF for winter instead to me suggests that the model did not apportion the FF fraction correctly.

We agree with the reviewer that the $BC_{FF}$ during winter (Fig. 3c) appears to have less pronounced (if any) bimodal behavior in comparison to $NO_X$ concentrations during winter (Fig. 3g). However, we would still claim that there are features in common. Both $BC_{FF}$ and $NO_X$ start with decreasing concentrations from 1:00-6:00, and then increase and stay on an elevated level until 22:00 when both concentrations show indications of decrease.

It is possible that the elevated precipitation during the winter (described in section 3.1) was responsible for increased wet deposition of BC while leaving the atmospheric $NO_X$ unaffected, resulting in blurring correlations between the two parameters during the winter.

We have added a few words to indicate that the bimodal pattern is stronger for the $NO_X$ concentrations compared to the $BC_{FF}$ concentrations.

We have checked our data again and found no reason why the calculations here should be invalid.

2. Page 11, line 25: If NOx is being oxidized in transport to the site, then shouldn't the CMFF and NOx show the highest correlation during winter when there is less photochemistry compared to the other seasons? Why is there such a better correlation in spring compared to the other seasons?

This is a very good question and hard to explain. As we answered on the former comment, one can speculate that increased wet deposition during winter resulted in a scavenging effect on BC while leaving the atmospheric $NO_X$ unaffected. Consequently, the correlations between BC and $NO_X$ during winter may have been weakened.

Studying the precipitation of the other seasons we find that the spring had the lowest precipitation, hence possibly explaining the improved correlation between $BC_{FF}$ and $NO_X$ during this period ($R^2$=0.41; p<0.001). However, the precipitation during spring is not significantly different from the precipitation during fall and summer (seasons with much lower $R^2$ value between $BC_{FF}$ and $NO_X$, $R^2$=0.07; p=0.021 and $R^2$=0.09; p=0.009, respectively). Hence, precipitation may only partially explain the increased correlation during spring.

3. Page 12, lines 1-3: I find it surprising that the CMwb was better correlated with NOx than CMFF. You explain this by stating that both NOx and CMwb have a seasonal dependence, but why would NOx and CMwb have the same seasonal dependence? Have you examined the correlation between CMwb and NOx to see if there is a change for the seasons?

$NO_X$ is mainly emitted from traffic, which is a rather stable emitter throughout the year. Furthermore, $NO_X$ has a longer lifetime during the cold/dark period of the year due to a lower rate of atmospheric photo-oxidation. Hence, $NO_X$ concentrations can be expected to be elevated (due to longer lifetime) during winter, and lower during summer. The same pattern applies for $CM_{WB}$ but for a different reason. Residents heat their homes through WB during winter, an activity that is almost absent during summer. We have clarified this in the text.

There are no changes in the correlation between $CM_{WB}$ and $NO_X$, all seasons show significant correlations.

4. Page 12, line 17. In Fig S6, why there is a very large spike in the concentration of levoglucosan in late March that is not observed in AAE (fig 2)? This sharp spike in levoglucosan suggests a biomass burning event, and I would have thought you would observe a corresponding increase in AAE if wood burning aerosols have a high AAE? Investigating the cause of the spike may help guide the choice of AAE for wood burning at the site.

The levoglucosan peak is derived from a 72h quartz filter with a stop-date of 2015-03-19. Hence, the filter represents ambient air during 16-18[th] of March 2015. This is indeed a pollution episode. The OC and EC concentrations are also elevated as displayed in Fig. 1a. Air mass trajectory analysis revealed that southeasterly (SE) air masses totally dominated (92 %) during this three day period. As pointed out in section 3.5, air masses from SE are associated to higher levels of aerosol loading. Higher aerosol concentrations from the SE are further supported by the study by Kristensson et al. (2008).

During this three day period we actually had somewhat elevated AAE (although it is hard to see in Fig. 2.). The mean AAE during this three day period was 1.44 (±0.03 standard deviation), which is

higher compared to the average AAE for the whole month of March 2015 (mean=1.37±0.09). However, since the measured $F^{14}C$ from the same filter showed a value of 0.86 there are some obvious contribution from fossil fuel combustion. Hence, we would not regard this pollution episode as being totally dominated by wood burning.

Nevertheless, there is a discrepancy in apportioned wood burning between the aethalometer model and the radiocarbon + levoglucosan method during this three day period. The aethalometer model apportions 54 % of the TC to wood burning while the radiocarbon + levoglucosan method apportions 90 % of the TC into wood burning. However, entangling the causes for this discrepancy is difficult. For instance, combustion of lignite has been shown to emit large quantities of levoglucosan, although it is a fossil source (Fabbri et al., 2008). Further, 55 % of the heat and power generation in Poland (located in the SE direction of Vavihill) are generated from lignite combustion (Burmistrz et al., 2016). Hence, deriving any source specific AAEs from this pollution episode should be conducted with great caution.

5. Section 3.5: did you use annual means for your comparison of the aethalometer and radiocarbon and levoglucosan apportionment? Did you see any changes in agreement between the two methods for the seasons, e.g. was there better agreement in summer or winter?

In section 3.4 we are comparing mainly annual means between the two source apportionment methods. As displayed in Fig. 6, we are comparing the methods on a seasonal basis. We re-analyzed possible differences on a seasonal basis through analysis of variance (ANOVA). There was a significant difference between the fossil fuel apportionments between the two methods for all seasons. There were no significant differences in apportionment of wood burning and biogenic carbonaceous aerosol between the two methods in any of the seasons.

We have clarified in section 3.4 that these comparisons discuss annual means.

6. Page13, line 15: In the aethalometer model, to calculate C1 and C2 only winter data was used as it was assumed that there would be negligible CMbio. However, the results from the radiocarbon and levoglucosan model suggest that biogenic carbon was not negligible during winter. I think that you should therefore include some discussion on how the presence of biogenic carbon in winter affected the source apportionment by the aethaolometer model.

As described in our answer to the first question, we have performed a sensitivity analysis where we increased or decreased the carbon in Eq. 10-11, while not changing the $b_{abs}$ parameters. This would be analogous to change the mass of non-light absorbing carbon. We have written a detailed discussion regarding this in section 3.4

7. Page 13, line 31: Why did you fix CMbio to -0.103 ug m-3 and not zero as you expect no CMBIO in winter?

In this case we are comparing our method (i.e. letting $CM_{Bio}$ vary outside the linear regressions) to the model proposed by Sandradewi et al. (2008) where they suggested solving a bilinear regression model with an allowed intercept. Hence, in order to adopt the Sandradewi method, we need to fix our intercept.

8. Section 3.6: My understanding is that in the proposed aethalometer model, the influence of biogenic carbon needs to be minimized in order to calculate C1 and C2. In addition to only using winter data, could you not also select data for the calculation by wind direction? My reading of this section is that there are geographically distinct areas around the sampling

site, and that filtering by wind direction you could further decrease in the biogenic influence (e.g. removing data when the wind is from a forested area or from the NE?) in the data.

This is a good idea suggested by the reviewer. In our study we have the lowest amount of incoming NE air mass during winter (i.e. 10 % of the air masses were from this direction). Hence, we believe that the selected winter data in our study exhibit favorable conditions in order to minimize biogenic carbonaceous aerosol.

We have added a sentence in section 3.5 describing the low abundance of NE air masses during winter.

9. Page 14, line 19-20: Why would SW air masses have high NOx but not be associated with carbonaceous aerosols, when traffic emissions are a significant source of both?

This is indeed very hard to explain. One explanation might be that the SW air masses are associated to increased precipitation ($R^2=0.19$; $p<0.01$). This correlation was particularly high during winter ($R^2=0.41$; $p<0.01$), a time when $NO_X$ can be expected to have increased lifetime due to low photochemical rates. Hence, it is possible that the increased SW-precipitation increased the wet deposition of carbonaceous aerosol particles while leaving the $NO_X$ unaffected.

We have added a sentence to offer this explanation in section 3.5.

References

Burmistrz, P., Kogut, K., Marczak, M., and Zwozdziak, J.: Lignites and subbituminous coals combustion in Polish power plants as a source of anthropogenic mercury emission, Fuel Process Technol, 152, 250-258, 10.1016/j.fuproc.2016.06.011, 2016.

Fabbri, D., Marynowski, L., Fabianska, M. J., Zaton, M., and Simoneit, B. R. T.: Levoglucosan and other cellulose markers in pyrolysates of miocene lignites: Geochemical and environmental implications, Environ Sci Technol, 42, 2957-2963, 10.1021/es7021472, 2008.

Kristensson, A., Dal Maso, M., Swietlicki, E., Hussein, T., Zhou, J., Kerminen, V. M., and Kulmala, M.: Characterization of new particle formation events at a background site in Southern Sweden: relation to air mass history, Tellus B, 60, 330-344, 10.1111/j.1600-0889.2008.00345.x, 2008.

---

## Author Comment (AC2) · 9 Mar 2017

General comments

The authors present an evaluation of the aethalometer model for carbonaceous particle source apportionment using radiocarbon and levoglucosan measurements, and quantify wood burning (WB) and fossil fuel combustion (FF) aerosol for a year-long dataset from a rural station in southern Sweden. The model is modified to allow for apportioned non-light absorbing biogenic aerosol to vary in time, improving the aethalometer-based source apportionment compared to radiocarbon and levoglucosan data. This is an interesting and solid study. The manuscript is very well written (with a few grammatical errors here and there, I suggest having it checked by a native English speaker), and the analysis is sound. I therefore recommend publication in ACP after the following comments have been addressed:

Specific comments

P. 4, l. 13: How efficient are the active carbon denuders? Please add information on that and the expected background. Is there any information on the evaporation of semi-volatiles from the particles after disturbance of the gas-particle equilibrium due to the denuders?

Genberg et al. (2011) conducted a full year source apportionment study at Vavihill 2008-2009. They conducted tests on the installed denuders and found 90-95% denuder efficiency (Genberg et al. 2011).

Further, when denuders were installed, Genberg et al. (2011) observed that obtained field blanks contained a carbon content similar to that of the sampled back filters, indicating that the possible negative artefact (due to disturbance of the gas-particle equilibrium) was low. Hence, Genberg et al. (2011) did not consider nor corrected for this artefact. In our study, we did not obtain any field blanks, however we have adopted the same approach as Genberg et al. (2011) since we are performing a similar study at the same measurement site with the same sampling setup, i.e. we have not corrected for this artefact.

We have added information on the efficiency of the denuders and the presumably low negative artefact caused by the denuders in section 2.1.

P. 6, l. 1 – 4: Why were the AAE values not calculated based on the actual data, or at least calculated and compared to literature data? Assuming an AAEFF of 1, and plotting/fitting babs vs wavelength (either averaged, or time-dependent, more appropriate here) can be used to derive AAEWB.

We do not see how we can calculate an $AAE_{WB}$ using the method suggested by the reviewer. To be able to derive $AAE_{WB}$ we need to know AAE values from the wood burning, i.e. through emission inventories. Our actual measured data shows the *babs* from a mixture of different aerosol sources. Hence, we cannot see how we could select any $AAE_{WB}$ based on these source mixtures of *babs*.

We instead obtained our source specific AAE values from emission inventories as displayed in Table 1. The mean $AAE_{WB}$ are in line with selected $AAE_{WB}$ values in previous aethalometer model source apportionment studies by Sandradewi et al. (2008) and Massabo et al. (2015). Further, our selected $AAE_{WB}$ value (1.81) is rather close to the recently suggested $AAE_{WB}$ (1.68) by Zotter et al. (2016).

P. 6, l. 11-12: If site specific sigma_abs were calculated based on linear regression of babs against EC, doesn't that imply an overestimation of sigma_abs, as light-absorbing OC is not included in EC? Please clarify.

This is true. We have added two sentences in order to clarify this.

P. 6, l.13: CM could also be SOA from WB and FF; should be mentioned here.

We have added this information.

P. 12, l. 2-3: This statement should be supported by references and more explanations.

We have developed and clarified this statement with some explanations.

P. 12, l. 29: Is the year-long time series correlated, or the diel evolution? I am assuming you are talking about the correlation of the time series. Apart from the non-optimal apportionment, a reason could also be, similar to the correlation of NOx and CMwb, a similar trend, but a different cause. CMFF is higher in winter than in summer, and so is NOx – potentially traffic emissions become more important in winter as well, or meteorological conditions favor the build-up of pollution episodes?

The correlation refers to the year-long time series since the levoglucosan data is in low time resolution (i.e. 72 h). This has been clarified.

This relation is hard to explain. It seems like the traffic emissions (judging from $CM_{FF}$ and $NO_X$ data) are relatively more important during winter compared to summer, at least in relation to TC. In absolute concentration, $CM_{FF}$ shows highest values during the fall and similar values during the other seasons (Table 4).

However, studying the relation on a seasonal basis we can see that a large portion of the correlation is explained by high correlations during winter and spring ($R^2$=0.77; p<0.001 and $R^2$=0.62; p<0.001, respectively) as compared to summer and fall ($R^2$=0.04; p=0.3 and $R^2$=0.35; p=0.001, respectively). Hence, one can speculate that the stronger correlation during winter and spring are associated to increased wood burning where some of the generated aerosols are absorbing light with a spectral dependence, AAE, close to 1, thus being falsely apportioned as fossil fuel combustion aerosol. It is also possible that lignite combustion aerosols from continental Europe, containing levoglucosan and exhibiting a spectral dependence of an AAE close to 1, may show higher abundance during these seasons, however we have no data supporting this speculation.

P. 13, l. 15: CMbio was assumed to be 0 for calculations of C1 and C2. Could this be a reason for the overestimation of CMFF?

We have added a new discussion paragraph were we performed a sensitivity analysis of the impact of non-light absorbing carbon on the aethalometer model results. It is very likely that the overestimation in CMFF can be explained by presence of non-light absorbing carbon (possibly biogenic carbon).

Technical comments
P. 7, l. 14: typo in chloride (also in subsequent use of trimethylsilyl chloride) P. 7, l.16: typo, should be dichloromethane.
This has been corrected.

References

Genberg, J., Hyder, M., Stenström, K., Bergström, R., Simpson, D., Fors, E. O., Jönsson, J. A., and Swietlicki, E.: Source apportionment of carbonaceous aerosol in southern Sweden, Atmos Chem Phys, 11, 11387-11400, 10.5194/acp-11-11387-2011, 2011.

Massabo, D., Caponi, L., Bernardoni, V., Bove, M. C., Brotto, P., Calzolai, G., Cassola, F., Chiari, M., Fedi, M. E., Fermo, P., Giannoni, M., Lucarelli, F., Nava, S., Piazzalunga, A., Valli, G., Vecchi, R., and Prati, P.: Multi-wavelength optical determination of black and brown carbon in atmospheric aerosols, Atmos Environ, 108, 1-12, 10.1016/j.atmosenv.2015.02.058, 2015.

Sandradewi, J., Prevot, A. S. H., Szidat, S., Perron, N., Alfarra, M. R., Lanz, V. A., Weingartner, E., and Baltensperger, U.: Using aerosol light absorption measurements for the quantitative determination of wood burning and traffic emission contributions to particulate matter, Environ Sci Technol, 42, 3316-3323, Doi 10.1021/Es702253m, 2008.

Zotter, P., Herich, H., Gysel, M., El-Haddad, I., Zhang, Y., Mocnik, G., Hüglin, C., Baltensperger, U., Szidat, S., and Prevot, A. S. H.: Evaluation of the Ångström exponents for traffic and wood burning in the Aethalometer based source apportionment using radiocarbon measurements of ambient aerosol, Atmos. Chem. Phys. Discuss., doi:10.5194/acp-2016-621, 2016.

---

## Author Response (AR2)

Dear editor,

We have clarified in the paragraph at the end of section 3.4 that the improved agreement between the methods refers mainly to the insignificant difference achieved in fossil fuel apportionment. Statistical information on this is written in the following sentence, i.e. "For the whole year, there would be no significant difference in apportioned fossil fuel carbon by the two methods (p=0.137)". We have further added information regarding the p-values of the insignificant differences in apportioned WB and biogenic carbon. Our changes are marked in yellow.

We would like to thank the editor for his careful reading and comments.

Comments to the Author:

[revised manuscript text omitted]